

**Approaching the diversity and density dilemma of the lebensspuren-tracemaker**
**tandem: a study case from abyssal Northwest Pacific**
Olmo Miguez-Salas[1,2*], Angelika Brandt[1,3], Henry Knauber[1] and Torben Riehl[1]
[1]Department of Marine Zoology, Senckenberg Research Institute, 60325 Frankfurt,
Germany.
[2]Departamento de Estratigrafía y Paleontología, Universidad de Granada, Av.
Fuentenueva 18002, Granada, Spain
[3]Department of Biological Sciences, Institute of Ecology, Evolution and Diversity,
Johann Wolfgang Goethe University Frankfurt, Max-von-Laue-Str. 13, Frankfurt, 60438,
Germany
*Corresponding author: olmo.miguez-salas@senckenberg.de
**Abstract**
In the deep-sea, the interaction between benthic fauna and substrate mainly occurs
through bioturbational processes which can be preserved as traces (i.e., lebensspuren).
Lebensspuren are common features of deep seafloor landscapes and usually more
abundant than the organism that produce them (i.e., tracemakers), rendering them
promising proxies to infer biodiversity. The density and diversity relationships between
lebensspuren and benthic fauna are to the present day unclear and contradicting
hypotheses have been proposed suggesting negative, positive, or even null correlations.
To test these hypotheses, in this study lebensspuren, tracemakers (specific epibenthic
fauna that produce these traces), degrading fauna (benthic fauna that can erase
lebensspuren), and fauna in general were characterized taxonomically at eight deep-sea
stations in the Kuril Kamchatka Trench area. No general correlation (over-all study area)
could be observed between diversities of lebensspuren, tracemakers, degrading fauna and
fauna. However, a diversity correlation was observed between specific stations, showing
both negative and positive correlations depending on: 1) the number of unknown



tracemakers (especially significant for dwelling lebensspuren); and 2) the lebensspuren
with multiple origins; and 3) tracemakers that can produce different lebensspuren.
Lebensspuren and faunal density were not correlated. However, lebensspuren density was
either positively or negatively correlated with tracemaker densities, depending on the
lebensspuren morphotypes. A positive correlation was observed for resting lebensspuren
(e.g., ophiuroid impressions, Actinaria circular impressions), while negative correlations
were observed for locomotion-feeding lebensspuren (e.g., echinoid trails). In conclusion,
lebensspuren diversity may be a good proxy for tracemaker biodiversity when the
lebensspuren-tracemaker tandem can be reliable characterized; and lebensspuren-density
correlations vary depending the specific lebensspuren residence time, tracemaker density
and associated behaviour (rate of movement), but on a global scale abiotic and other biotic
factors may also play an important role.
**Introduction**
Neoichnology studies the interactions between animals and substrates (i.e., bioturbation
processes) in modern environments as well as their final products, the so-called
lebensspuren (German for "life traces"; e.g., faecal casts, trails, mounds, burrows) (Ewing
and Davis, 1967; Gage and Tyler, 1991). Lebensspuren are highly precise portraits of the
diverse linkages between environmental conditions and the animal responses to them.
Thus, neoichnological analysis provides a useful tool set to infer environmental factors
not only in contemporary environments but also deliver evidences to past environments
through comparison between lebensspuren and trace fossils (Buatois and Mángano,
2011). However, neoichnology as a field is not yet as developed as paleoichnology (i.e.,
trace fossil research), and most quantitative studies are restricted to shallow marine
environments and tank experiments (e.g., shoreface, foreshore, marginal marine settings)
(La Croix et al., 2022 and references therein). Even though the abyssal zone (i.e., 3500-



6500 m deep) represents the largest marine ecosystem and covers approx. 75% of the
seafloor (Ramirez-Llodra et al., 2010; Watling et al., 2013), neoichnological analyses are
scarce and limited, mainly due to the cost of observation and sampling procedures (e.g.,
Heezen and Hollister, 1971; Przeslawski et al., 2012; Bell et al., 2013; Miguez-Salas et
al., 2022). Thus, neoichnological analyses emerge as a promising tool to enhance our
understanding of deep-sea environments and faunal-sediment interactions.
Diversity and density analyses are two main components of quantitative marine
ecological research (Halpern and Warner, 2002). Deep-sea neoichnological studies have
addressed diversity and density characterizations by considering all identified
lebensspuren morphotypes as "species" (Przeslawski et al., 2012; Bell et al., 2013).
However, tracemaker (i.e., the benthic organisms that produce the observed lebensspuren)
diversity and density have been approached from a generalist perspective as megafauna,
epifauna, or lebensspuren-forming epifauna (Young et al., 1985; Dundas and Przeslawski,
2009; Przeslawski et al., 2012; Bell et al., 2013).
Early deep-sea neoichnological studies suggested that lebensspuren diversity is
proportional to faunal diversity (Kitchell et al., 1978, Young et al., 1985). However, more
recent studies have shown no correlation between epifaunal and lebensspuren richness
(Przeslawski et al., 2012) and that lebensspuren diversity was not similar to that of
epifaunal lebensspuren-forming diversity (Bell et al., 2013). Bell et al. (2013) stated that
"improvements in imaging technology allow more refined classification of lebensspuren
and species, which may affect the strength of the correlation between faunal and
lebensspuren diversity, compared with the more direct proportionality of faunal and
lebensspuren diversity demonstrated in earlier studies". Thus, in deep-sea research,
diversity comparisons based on more precise taxonomic tracemaker identification and



differentiation are a pending task, promising a deeper understanding of the dependencies
between fauna and lebensspuren variability.
In the case of lebensspuren density, early studies revealed an inverse relationship with
faunal density (Kitchell et al., 1978, Young et al., 1985; Gerino et al., 1995). These studies
suggested that this relationship is related to the fact that lebensspuren formed in low
biomass regions have the capacity to persist for a long time (high residence time),
ultimately leading to a steady increase of the lebensspuren density through accumulation.
Nevertheless, recent data seemed to conflict with this initial assumption. Przeslawski et
al. (2012) observed that lebensspuren and epifaunal abundance do not have any
relationship; and, contrastingly, Bell et al. (2013) found a strong positive relationship
between lebensspuren and faunal densities (see Fig. 10 in Bell et al., 2013). These newer
results show that megafaunal activity may not be the only significant factor for
lebensspuren destruction or preservation. Small scale biotic factors (e.g., microbial
degradation), as well as abiotic factors (e.g., hydrodynamic regimes, sedimentations rates,
sediment composition) may limit lebensspuren residence time and density (Wheatcroft et
al., 1989; Smith et al., 2005; Miguez-Salas et al., 2020). In summary, strong variability
in the few previous studies and conflicting conclusions drawn from these highlight that
neoichnology and its fundamental concepts are still in their infancy.
Despite the presence of many lebensspuren on the deep seafloor (Heezen and Hollister,
1971), very few organisms are recognized in the process of forming these features. Thus,
understanding the density-diversity relationship between lebensspuren and benthic
megafauna may help decipher variability of the former indirectly (i.e., without having
seen the organisms). The research presented here aims to compare diversities indices and
densities of lebensspuren, specific tracemakers, and megabenthic fauna from the
Northwest Pacific Abyssal Plain in the direct vicinity of the Kuril Kamchatka Trench



(KKT) (Fig. 1). By conducting a detailed classification of both lebensspuren and
tracemakers, this research wants to go one step further with the main objective to test
previous diversity and density hypotheses about the relationship between the variability
of lebensspuren and fauna. The geographic region is well-studied as it has a long research
history that began with eleven expeditions onboard of R/V *Vityaz* (Russian expeditions;
1949, 1953 and 1966) and was further extended during recent campaigns with R/V *Sonne*
(German-Russian expeditions; KuramBio I (2012) and KuramBio II (2016)). All of these
expeditions resulted in one of the best taxonomic baseline of the fauna (e.g., Zenkevitch
et al., 1955, Zenkevitch, 1963; Belyaev, 1983; Brandt and Malyutina, 2015; Brandt et al.,
2020; Saeedi and Brandt, 2020 among others).
**Material and methods**
*Study sites, data acquisition, and video analysis*
The joint German–Russian expedition KuramBio 1 (Kurile Kamchatka Biodiversity
Studies) on board of the RV *Sonne* (cruise SO223) to the Kuril–Kamchatka Trench and
adjacent abyssal plain took place between July 21st and September 07th 2012 (Brandt
and Malyutina, 2012). During the expedition, 13 Ocean Floor Observation System
(OFOS) deployments were conducted (Table 1) to study eleven deep-sea stations between
34º–48ºN and 147º–157ºE (Fig. 1) with video cameras. Stations 3 and 4 were located at
the upper slope of the KKT, and stations 1, 2, and 5–11 in the adjacent abyssal plains
(Fig. 1). The depths of the stations ranged from 4,868 m to 5,768 m.
The OFOS was lowered into the water at the CTD position. The first 300 meters lowering
was conducted with 0.5 m/s, and then the speed was increased to 0.8 m/s while the ship
was kept in position. At 500 meters above ground, the speed was reduced to 0.5 m/s, and
further reduced to 0.3 m/s at 200 meters above ground. As soon as visual contact with the
bottom was established, the winch was stopped. The ship started moving with 0.5 knots



above ground in the appropriate direction, which was chosen depending on the current
and wind situation. Then, the winch operator manually kept the OFOS at an appropriate
distance from the seafloor to observe the seafloor benthos. Two laser pointers having a
distance of 10 cm between each other were used as a scale. The first four deployments
were aborted due to technical problems, affecting stations 1–3 (Table 1). Thus, limited
video footage was obtained. Moreover, station 7 has no HD video (i.e., this station is not
considered for the current analysis). All technical work, including preparation before and
caretaking after (including video download) the deployment was conducted by the
scientific-technical service ("WTD", Wissenschaftlich-Technischer Dienst, Jörg Leptien,
Reederei).
At each station, still images were extracted from the OFOS videos at a rate of one frame
per five seconds. These still images were subsequently further sub-sampled to delete
frames that were out of focus — as the rolling of the ship in the ocean swell resulted in
an up and down movement of the OFOS — and to reduce overlap between frames. Then,
randomly selected frames per station were studied (400 still images in total), covering
a seafloor area of 878 $m^2$ (109 $m^2$ per station approx.). These still images were uploaded
to the BIIGLE 2.0 software for later annotation and measurements (Langenkämper et al.,
2017). Specific frames were treated with Fiji software (Schindelin et al., 2012) to enhance
the visibility (CLAHE tool) of certain lebensspuren features (Miguez-Salas et al., 2019).
*Lebensspuren classification and tracemaker identification*
Lebensspuren morphotypes were categorized in terms of inferred tracemaker behaviour
during the construction, morphology, and tracemaker taxonomic origin. The behavioural
classification was adapted from Seilacher's (1954) categories for marine lebensspuren: i.
Resting (imprints of stationary animals); ii. Locomotion-feeding (sediment displaced by
the movement of deposit feeders and surface sediment disturbances formed as organisms



are foraging); iii. Wasting (e.g., faecal casts, pellets); and iv. Dwelling (e.g., mounds and
burrows). Morphological features measured included in the classification were length,
width, and diameter. Lebensspuren with unclear morphology and origin (e.g., degraded
faecal casts, trails with diffuse outlines) were not considered in this study. Also, as the
resolution of the still images is below high-definition (<1280x720 pixels) lebensspuren
and fauna smaller than 1 cm (macrofauna and smaller) have not been considered in this
study. Hence, this study focusses on megafauna (i.e., fauna > 1 cm) which is implied
throughout this study when fauna is mentioned from hereon.
Open nomenclature has been used for megafauna taxonomic identification following the
recommendations for image-based identifications proposed by Horton et al. (2021). All
differentiated morphotypes are henceforward referred to as "species" for simplicity.
Then, fauna has been grouped into different categories for comparisons with the diversity
and density of lebensspuren: 1) tracemakers (fauna that has been clearly recognized as
maker of a trace); 2) degrading fauna (fauna that can affect lebensspuren density
negatively by eroding the seafloor); and 3) benthic fauna (all fauna identified in the still
images).
*Statistical analysis*
For statistical analysis, all identified lebensspuren and fauna morphotype were treated as
"species". Diversity indices (Shannon–Wiener H′ ($\log_e$) and Simpson's D) and evenness
(J′) were calculated for the four groups: lebensspuren, tracemaker fauna, degrading fauna,
and fauna. As the data from all groups show non-parametric distribution throughout all
stations, diversity variability among stations was tested using Wilcoxon signed-rank test
(considering all groups and all indices). Then, the Spearman rank correlation was used to
test the relationships between the diversity indices of all groups.



For density correlations (Spearman rank correlation), since the number of frames was the
same (i.e., same observation area), the analyses were conducted considering the total
density per station of all groups individually. Additionally, lebensspuren and tracemakers
densities were subdivided into wasting, resting and locomotion-feeding (dwelling was not
considered because the tracemakers of most dwelling lebensspuren are unknown).
To investigate potential differences within the four groups (lebensspuren, tracemakers,
degrading fauna, and total benthic fauna) between stations, multivariate analysis was
conducted. First, a square root transformation was carried out to give less weight to the
more abundant species and lebensspuren. Then, differences in the composition of the four
groups between stations were assessed using hierarchical cluster analysis and displayed
as non-metric multidimensional scaling plots (n-MDS). Both plots were constructed using
the Bray–Curtis similarity index. All statistical procedures were conducted using PAST
v. 4.12 (Hammer, 2001).
**Results**
A total of 9,426 lebensspuren were identified and classified from 400 still images,
corresponding to 23 morphotypes associated with dwelling, wasting, resting, and
locomotion-feeding behaviours (Fig. 2; Table 2). The fauna comprised a total of 4,009
individual animals that were classified into 93 different species, of which 66 were
classified as degrading fauna and 43 as tracemakers (with 790 and 676 individuals
respectively) (Table 3; Supplementary file 1). Linking dwelling lebensspuren with
tracemakers was mostly impossible except for rare and ambiguous cases where
vermiform organisms, most likely polychaetes, partially emerged from paired burrows
(Fig. 2P). Tracemaker identification was possible in the majority of the cases for wasting
lebensspuren, however, it is common that different tracemakers produce the same
lebensspuren morphotypes and that several morphotypes of lebensspuren are produced



by one tracemaker species (see Table 2). However, in the case of cf. *Elpidia* — the most
abundant tracemaker of station 4 (see Supplementary file 1) — the complete
characterization of its associated rounded faecal cast (smaller than 1 cm) was impossible
due to image resolution limitations. Tracemaker identification of locomotion-feeding
lebensspuren was mostly possible except for mounded trails which have been produced
by endobenthic organisms. However, as for wasting lebensspuren, also in this case
different tracemakers can be responsible for similar trails (see Table 2). Tracemaker
identification of resting lebensspuren has been possible in most cases.
The Wilcoxon signed-rank test revealed that for all groups the median diversity was
significantly different between stations, being lower at stations 9 and 11 (Fig. 3).
Moreover, faunal diversity showed a standard deviation three orders smaller than the
values reported for lebensspuren, tracemakers, and degrading fauna. Lebensspuren
diversity indices (Shannon–Wiener, Simpson's and Evenness) of the over-all KKT area
(considering all the eight stations together) showed no correlation with the other three
groups (tracemakers, degrading fauna, and benthic fauna). The only strong diversity
correlation resulting from the Spearman rank analysis was between tracemakers and
degrading fauna ($R^2 > 0.88$, $p < 0.01$).
The density correlation matrix revealed no significant correlation between the fauna and
the other groups (see Fig. 4). The degrading fauna showed a positive correlation with
tracemaker and wasting tracemakers densities. Also, tracemakers and wasting
tracemakers densities are positively correlated (Fig. 4). In case of the lebensspuren data,
a positive density correlation was obtained between lebensspuren and wasting
lebensspuren as well as resting lebensspuren and resting tracemakers while a negative
correlation was observed for locomotion-feeding lebensspuren and their tracemakers
(Fig. 4).





Inter-station similarity of lebensspuren assemblage composition was generally high (Fig.
5 A), ranging from 75–82% similarity in the cluster analysis. The n-MDS showed that
lebensspuren assemblages from stations 5, 6, 8, and 10 are different from the trench
(stations 3 and 4) and the southern stations (stations 9 and 11) (Fig. 6 A). The southern
stations were less diverse, similar (82% similarity; Fig. 5A) and dominated by rounded
faecal casts produced by *Scotoplanes* spp. The trench stations were characterised by
diverse and slightly less similar assemblages (75% of similarity) dominated by dwelling
lebensspuren (e.g., paired, lined or cluster burrows), knotted faecal casts (*Peniagone*
spp.), ophiuroid impressions (Ophiuroidea), circular impressions (Actinaria) and thick M-
trails (Asteoridea and *Echinocrepis* spp.). Stations 5, 6, 8, and 10 showed diverse
lebensspuren assemblages dominated by smooth (cf. *Benthodytes*, *Psychropotidae*) and
coiled faecal casts (*Psychropotidae*), rosette-shaped traces and thick flat trails
(Asteoridea, cf. *Benthodytes*, *Psychropotidae*) (Fig. 6 A).
The hierarchical cluster diagram for tracemakers, degrading fauna and fauna showed less
similarity between stations than it was the case for lebensspuren, especially for
tracemakers and degrading fauna (values ranging from 20–55% similarity in the cluster
analysis) (Fig. 5 B–D). However, the trench stations (Stations 3 and 4) and the southern
stations (Stations 9 and 11) seemed to have similar compositions respectively. The low
inter-station similarity of tracemakers, degrading fauna and fauna assemblages was also
reflected in the n-MDS plots where the spacing between stations was considerably higher
than in the lebensspuren plot (Fig. 6 B–D).
**Discussion**
The obtained results from the KKT area reveal that the relationship between lebensspuren,
tracemakers, and fauna may be more complicated than previously hypothesized. On the
one hand, a general null diversity correlation has been observed between lebensspuren,



tracemakers and fauna. On the other hand, density correlations seem to be morphospecific
(e.g., depending on the lebensspuren-associated behaviour). But to what extent do the
obtained results contradict or corroborate previous results and what are the limitations
when addressing the diversity and density of lebensspuren?
*Fauna, tracemakers and lebensspuren diversity: a complex relationship*
Previous comparisons between lebensspuren and faunal diversity have given rise to
different contrasting hypotheses. Pioneering research showed positive correlations (e.g.,
Kitchell et al., 1978, Young et al., 1985). Later on, several studies showed no correlation
at all (e.g., Tilot, 1995; Turnewitsch et al., 2000; Przeslawski et al., 2012). All these
studies have in common that the diversity comparison was addressed from a general
perspective, especially for tracemaker organisms. Comparisons were done either
considering megafaunal species (Young et al., 1985), epifaunal species (Przeslawski et
al., 2012) or certain taxonomic groups of organisms (e.g., fish, holothurians, crinoids;
Kitchell et al., 1978). Only Bell et al. (2013) approached the comparison between
lebensspuren and fauna in greater detail considering groups of lebensspuren-forming
epifauna and using indices to quantify lebensspuren diversity (e.g., Simpson, Shannon-
Wiener), discovering that "Lebensspuren diversity was generally high and not similar to
that of lebensspuren-forming faunal diversity". However, the links between specific
tracemakers and their lebensspuren and the subsequent tracemaker diversity indexes are
missing in Bell et al.'s (2013) study. In this study we have tried to close this knowledge
gap by comparing the lebensspuren diversity with not only the faunal diversity but also
the tracemaker and degrading fauna (i.e., fauna that may alter the lebensspuren
assemblage by erosion/degradation). Our results show that lebensspuren diversity
(Simpson, Shannon-Wiener, and Evenness) is not related to fauna, tracemaker or
degrading fauna diversity. This finding seems to corroborate the latest results of a non-





existent correlation (Przeslawski et al., 2012; Bell et al., 2013), but can this lack of
correlation be expected in all deep-see settings?
Before answering this question, the limitations of quantifying deep-sea lebensspuren
diversity should be considered. There are several problems when it comes to quantifying
lebensspuren diversity (e.g., image resolution, camera systems, unknown tracemakers,
observation scale, trace degradation), but the most important is linked to their genesis. In
other words, the same lebensspuren morphotypes (or indistinguishable lebensspuren) can
be produced by different tracemakers and one tracemaker can produce different
lebensspuren (see Table 2). For example, in case of this study, smooth faecal casts could
have been produced by different holothurians (e.g., cf. *Pseudostichopus*, *Psychropotes*,
*Synallactidae*, *Benthodytes*) and *Psychropotes* can be linked to the production of coiled
and smooth faecal casts as well as thick flat trails (Fig. 2G). Thus, when comparing their
diversity, the basis that each lebensspuren morphotypes may not be related to one specific
species and *vice versa*, should be considered. However, the fact that in our study general
lebensspuren diversity did not correlate with tracemaker diversity does not mean that this
will be the case in all deep-sea settings if the tracemaker-lebensspuren tandem can be
characterised more precisely or tracemakers produce just one specific lebensspuren
morphotypes.
In our study area, different correlations between tracemakers and lebensspuren could be
observed when comparing the diversity among specific stations. For example, when
restricting the comparison to the southern stations (stations 9 and 11), a correlation was
observed between Simpson and Shannon-Wiener indexes of tracemakers and
lebensspuren (Fig. 3). This was due to the fact that the assemblage is dominated by traces
for which we have been able to identify the tracemakers (e.g., rounded faecal casts of
*Scotoplanes*). On the contrary, when focussing on the trench stations (stations 3 and 4), a



negative correlation could be observed between Simpson and Shannon-Wiener indexes
of tracemaker and lebensspuren diversities (Fig. 3). This could be attributed to the
relatively large gap in our data regarding the origin of most traces of the lebensspuren
assemblage (stations 3 and 4 have a high abundance of dwelling lebensspuren (see Table
3); single burrows, mounds, cluster burrows for which tracemakers are unknown) and
dominant tracemakers (*Elpidia*) whose traces cannot be correctly quantified due to image
resolution limitations (small rounded faecal casts).
The enhancement of image resolution and the increase of deep-seafloor area covered by
still image surveys may allow to improve lebensspuren classification and their tracemaker
identification. There is a lot of room for improvement, especially with regard to
locomotion and feeding lebensspuren. High definition still images will allow to
characterize, for example, small morphological features of trails (e.g., podia marks from
asteroids, echinoid spine impressions), allowing for a much more detailed classification
than what could be achieved for this study. In the case of dwelling lebensspuren diversity
comparison is significantly more complicated because trace morphology is largely hidden
below the seafloor surface, reducing the possibility to differentiate between different
burrow morphologies while tracemakers are mostly unknown due to a predominantly
endobenthic lifestyle (e.g., Brandt et al., 2023). Furthermore, burrows and other dwelling
lebensspuren also could potentially have multiple origins (e.g., a paired burrow can be
produced by multiple species of polychaetes or bivalves).
The fact that the same lebensspuren morphotypes can be produced by different
tracemakers and one tracemakers can produce different lebensspuren will affect the
establishment of a positive or negative diversity correlation. Also, the existence of
unknown tracemakers will contribute to the correlation variability. However, as the
obtained results show in specific stations, when the assemblage is dominated by traces



with identifiable tracemakers, lebensspuren analysis emerges as a promising tool to
predict tracemaker diversity. Despite of these optimistic results, it is fair to say that much
more research is needed — with high definition surveys (e.g., videos, images) — to close
existent knowledge gaps in the lebensspuren-tracemaker tandem. Moreover, we
emphasize that when using lebensspuren as a proxy for biodiversity, the diversity
correlation should be made between lebensspuren and tracemakers, rather than with
overall benthic fauna as no correlation has been observed in case of comparison with the
latter.
*Tracemaker and lebensspuren density: morphospecific relationship*
The density comparisons between lebensspuren, degrading fauna and total fauna revealed
no correlation, similar to previous research (Przeslawski et al., 2012). However, when
comparing lebensspuren and tracemakers a positive and negative correlation can be
observed (Fig. 4). The density of locomotion-feeding lebensspuren is inversely correlated
with their tracemaker density while resting lebensspuren are positively correlated with
their tracemakers densities. These group-specific correlations conflict with previous
research that showed generally positive (e.g., Bell et al., 2013) or generally negative
density correlations (e.g., Kitchell et al., 1978; Young et al., 1985). The difference with
these previous studies may be due to the fact that their density comparisons considered
the total fauna instead of separate functional groups (see Fig. 10 in Bell et al., 2013), not
considering their specific impact on the sediment.
Trace residence time is the period during which a trace is recognizable on the sea floor
before it is destroyed (Wheatcroft et al., 1989). It is commonly accepted that lebensspuren
density values reflect the balance between lebensspuren formation and lebensspuren
destruction/degradation either by biotic (e.g., microbial degradation, degrading fauna,
epifaunal rate of movement) or abiotic factors (e.g., hydrodynamics, burial) (Wheatcroft



et al., 1989). However, not all lebensspuren have the same residence time. Thus, traces
not actively maintained by animals are usually ephemeral features with lifespans of days
to weeks (e.g., faecal casts can be degraded within 1-2 weeks; Smith et al., 2005) while
locomotion-feeding and resting lebensspuren have higher residence time as they are
impressions on the seafloor (see Fig. 8 in Wheatcroft et al., 1989 or Fig. 5 in Miguez-
Salas et al., 2020). Very little is known about the residence time of dwelling lebensspuren,
some tracemakers live inside them for their whole life while others change several times
and their burrows get passively filled. Thus a wide range of residence times may be
expected. However, in any of the cases since the sedimentation rate is usually low in the
deep-sea, dwelling lebensspuren should have higher residence time than wasting
lebensspuren and similar or higher than locomotion-feeding and resting lebensspuren.
In case of this study the density of locomotion-feeding lebensspuren (e.g., thick M-trails),
on the one hand, was inversely correlated with tracemaker density. This could be for two
reasons: 1) a high residence time of these lebensspuren while the respective tracemakers
may no longer be in the study area; and 2) these lebensspuren represent a foraging
behaviour in which the tracemakers tend to continuously search the seabed for food, often
over a wide area (i.e., high rate of movement). Thus, a large quantity of lebensspuren may
be produced by a single individual tracemaker in continuous movement. The density of
resting lebensspuren (e.g., circular impressions, asteroid impressions), on the other hand,
was in this study directly correlated with tracemaker density. This is not surprising
because even though these lebensspuren have a high residence time, their tracemakers
(e.g., asteroids, actiniarians) have low rates of movement (Durden et al., 2015; 2019). In
such cases, a high density of resting lebensspuren should always be linked to a high
density of their tracemakers.





The density correlation between wasting lebensspuren and their tracemakers showed a
slightly positive but not significant correlation (Fig. 4). Maybe this is due to the fact that
in some cases we were not able to quantify the exact number of faecal casts. For example,
in station 4, the lebensspuren of the dominant tracemakers (*Elpidia*; more than 150
specimens were identified) were not correctly quantified due to image resolution
limitations (small rounded faecal casts). Thus, presumably a positive density correlation
between wasting lebensspuren and their tracemakers should be expected. However, this
assumption may be disturbed by their tracemakers behaviours since their feeding activity
can be expected to depend on grain size, availability and quality of the nutrients among
other environmental factors (e.g., Jumars and Wheatcroft 1989; Ginger et al., 2001).
The observed variability in the lebensspuren density correlations show a complex
scenario even without considering biotic and abiotic factors that cannot be characterized
through still images. For example, it has been demonstrated that meiofauna and
microfauna have the ability to smoothen and eventually fully erase surficial biogenic
structures through small scale, grain-by-grain jostling of particles (e.g., Cullen, 1973).
These "small" biotic processes are impossible to quantify through images, however, it has
to be kept in mind that these will have affected also the lebensspuren density that we
quantified for this study. Moreover, previous studies assumed that abiotic lebensspuren
degradation rates are constant over the lebensspuren residence time period, but recent
studies show that this may not be always true (Miguez-Salas et al., 2020). The effects of
abiotic factors on the density of the studied assemblages as well as those of some biotic
factors (e.g., microbial degradation which cannot be characterized in a still image) are out
of the scope of this research but should be considered in future studies and need to be
kept in mind when interpreting seafloor images.



**Conclusions**

The neoichnological analysis of the KKT area reveals a general null diversity correlation between lebensspuren, tracemakers and fauna while density correlations vary depending on the lebensspuren morphotypes. The further conclusions of this study are:

The fact that the same lebensspuren morphotypes can be produced by different tracemakers and one tracemakers can produce different lebensspuren will affect the establishment of a positive or negative diversity correlation.

The existence of unknown tracemakers will contribute to the diversity correlation variability. However, lebensspuren diversity may be a good proxy for tracemaker biodiversity when the lebensspuren-tracemaker tandem can be reliable characterized.

Lebensspuren density can be positively or negatively correlated with tracemaker densities depending on the specific lebensspuren residence time and tracemaker behaviour (e.g., locomotion, resting).

Lebensspuren-density correlations may be control on a global scale by abiotic (e.g., hydrodynamics, grain size, organic matter) and biotic factors (e.g., microbial degradation).

**Acknowledgments**

Special thanks to the German Federal Ministry of Education and Research (BMBF) for funding this project (PTJ, Grant 03G0223A to A. Brandt). We also thank the crew of *R.V. Sonne*. We thank the Russian coordinator of the expedition M. Malyutina. The research of O. Miguez-Salas was funded by a Humboldt Postdoctoral Fellowship from the Humboldt Foundation. This is contribution #7 of the Senckenberg Ocean Species Alliance (SOSA).

**Author's contributions**



O.M.S., T.R., performed the data acquisition and treatment. O.M.S., T.R., and A.B., wrote
and designed the main manuscript text. O.M.S., H.K., prepared all figures, tables, and
supplementary material. All authors reviewed and edited the manuscript at multiple stages
and approved it for submission.

**Availability of materials and data**
All data generated or analysed during this study are included in this published article. The
raw data used for this study is in the Supplementary Information file.

**Competing interests**
The authors declare no competing interests

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





**Figure captions**
**Fig. 1** Map of the study area (Kuril-Kamchatka Trench area) and the location of the
analyzed deep-sea stations.

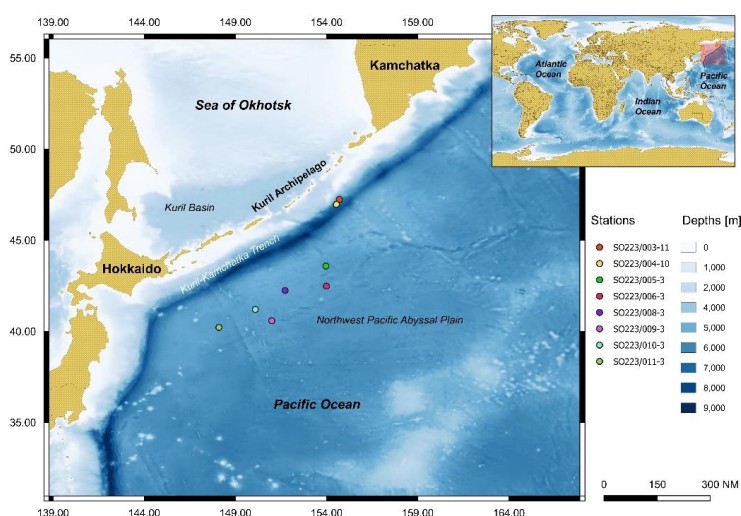


**Fig. 2** Examples of lebensspuren morphotypes observed and quantified in this study. A)
Thick M-trail produced by Asteroidea. fam. gen. sp. 1; B) Mounded trail (unknown
tracemaker); C) Thin flat trail (balck arrow; unknown tracemaker) and rounded faecal
casts (white arrow) produced by *Scotoplanes* sp. 1; D) Thick M-trail produced by
Echinoidea. fam. gen. sp. 5; E) Wavy faecal cast produced by *Peniagone* sp.1 to
*Peniagone* sp. 3; F) Knotted faecal cast produced by *Peniagone* sp.1 to *Peniagone* sp. 3;
G) Coiled (white arrow) and smooth (black arrow) faecal cast produced by *Psychropotes*
morphospecies 2; H) Smooth (black arrow) faecal cast produced by various tracemakers
(see Table 2); I) Rosette-shape trace (white arrow) produced by an echiuran worm and
mound shape nearby (black arrow); J) Spirals faecal cast produced by Enteropneusta gen.
sp. 1; K) Switchbacks faecal cast produce by Torquaratoridae. gen. sp. 1; L) Circular
impression produce by Actiniaria. fam. gen. sp. 1; M) Asteroid impression produced by
an Asteoridea (Asteroidea. fam. gen. sp. 3, 4, 7, 8, 9); N) Mound (white arrow) with a
semi-buried asteroidean nearby (black arrow); O) Single burrow located in the apex of a
cone-shaped mound; P) Paired burrow with an unidentified organism coming out; Q)
Three paired burrows; R) Cluster burrows; S) Lined burrows (black arrow).

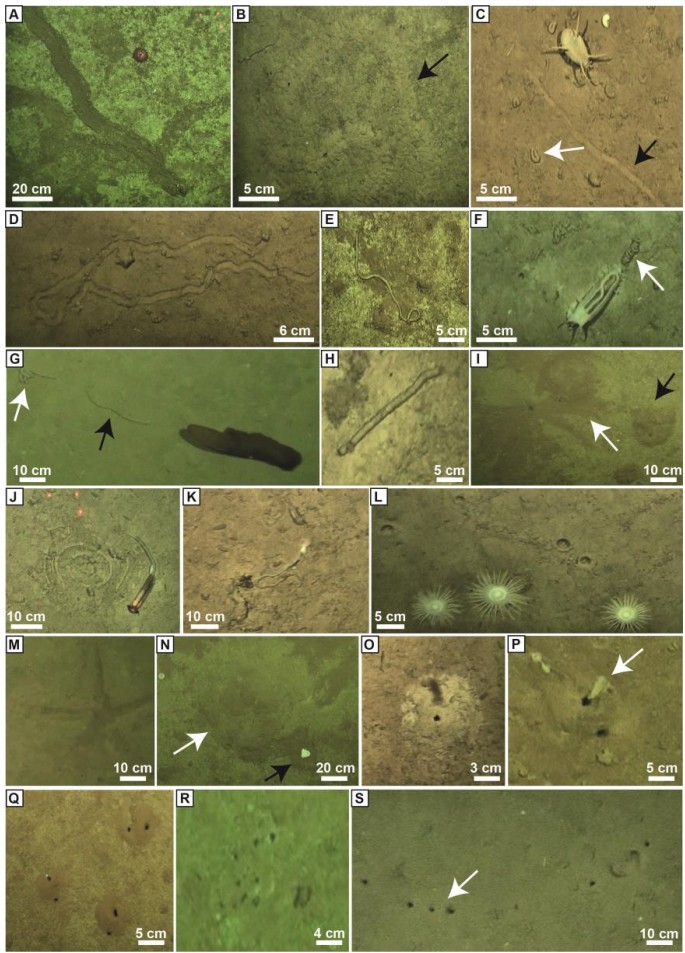

**Fig. 3** Comparison of median diversity indices (Simpson's, Shannon–Wiener and Evenness) of lebensspuren, tracemakers, degrading fauna and fauna at each station. Each lebensspuren morphotypes was considered a different species for calculations.


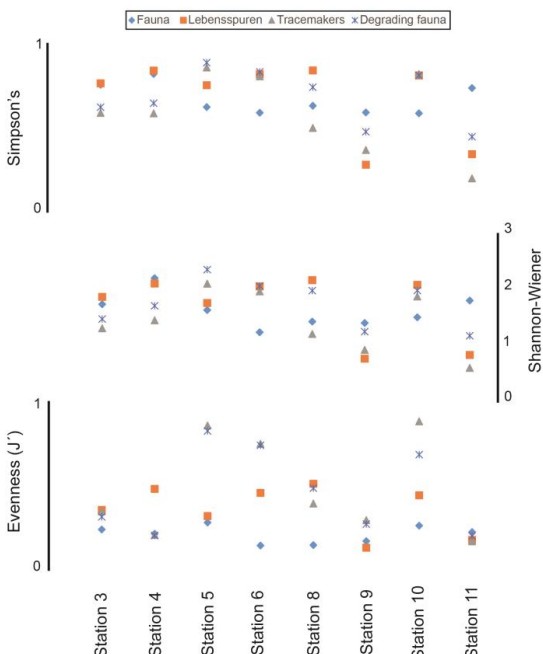

**Fig. 4** Density correlation matrix for lebensspuren, tracemakers, degrading fauna and fauna. Lebensspuren and tracemakers densities were subdivide into wasting, resting and locomotion-feeding (dwelling was not considered since the tracemakers of most dwelling lebensspuren are unknown). Boxed dots indicate correlations where p < 0.05.

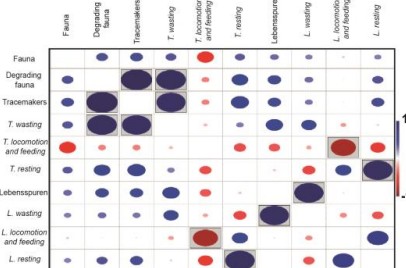

**Fig. 5** Hierarchical cluster diagram (constructed with Bray-Curtis similarity matrix) of the abundances of lebensspuren (A), tracemakers (B), degrading fauna (C) and fauna (D) at each station.



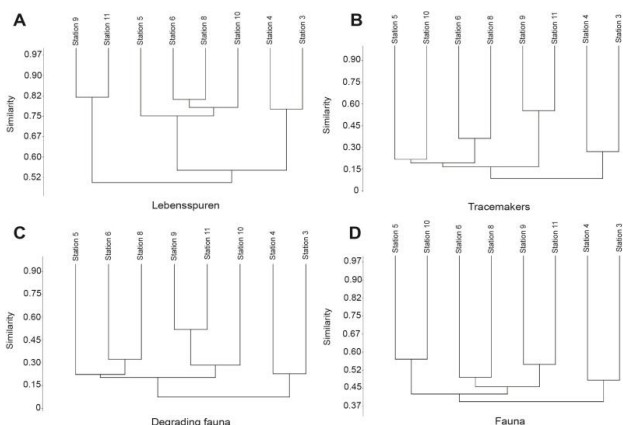


**Fig. 6** Multivariate similarity represented with a non-metric multidimensional scaling (n-
MDS) plots of lebensspuren (A), tracemakers (B), degrading fauna (C) and fauna (D) at
each station. Note that the only plot that stations are together is for lebensspuren
abundance.

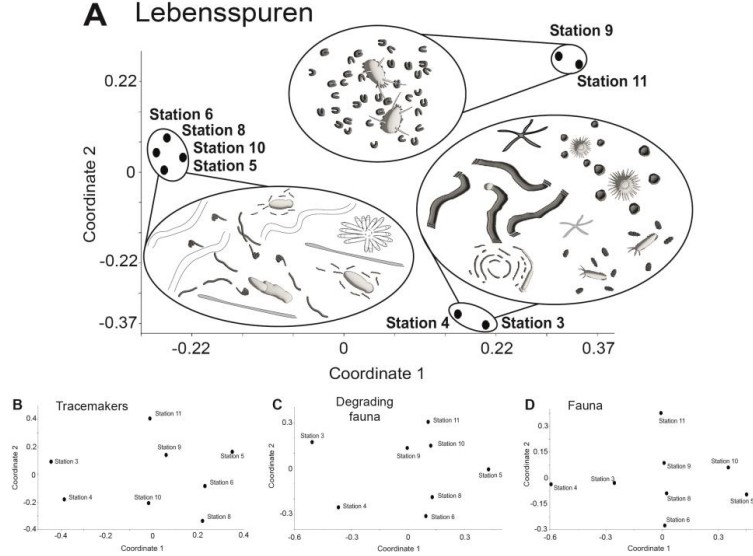







**Table captions**
**Table 1.** Station data of the OFOS deployments during KuramBio (2012). "Start" and
"End" coordinates refer to the time between bottom view and beginning of heaving
(survey duration). Notes: The first four deployments were aborted due to technical
problems.

| Station | Start Date | Start | End | Depth (m) | Survey duration /min | Notes |
|---|---|---|---|---|---|---|
| 01-03 | 28.07.2012 | 44°0.03' N 157°18.52' E | 44°0.01' N 157°18.50' E | 5315-5312 | 7 | Not enough good frames |
| 01-09 | 30.07.2012 | - | - | - | - | No video |
| 02-03 | 01.08.2012 | 46°14.04' N 155°33.05 E | 46°14.04' N 155°33.05' E | 4868-4868 | 4 | Not enough good frames |
| 03-03 | 04.08.2012 | - | - | - | - | No video |
| 03-11 | 06.08.2012 | 47°14.31' N 154°42.35' E | 47°13.80' N 154°43.16' E | 4990-5073 | 75 | |
| 04-10 | 08.08.2012 | 46°58.00' N 154°32.48' E | 46°58.48' N 154°31.44' E | 5768-5591 | 152 | |
| 05-3 | 09.08.2012 | 43°35.03' N 153°57.95' E | 43°34.64' N 153°58.60' E | 5377-5374 | 125 | |
| 06-3 | 13.08.2012 | 42°28.97' N 153°59.91' E | 42°28.18' N 153°59.90' E | 5298-5308 | 81 | |
| 07-3 | 16.08.2012 | 43°2.23' N 152°59.16' E | 43°1.81' N 152°59.70' E | 5222-5221 | 71 | Video with not enough definition |
| 08-3 | 19.08.2012 | 42°14.61' N 151°43.50' E | 42°14.42' N 151°42.91' E | 5125-5125 | 61 | |
| 09-3 | 22.08.2012 | 40°34.99' N 151°0.03' E | 40°34.47' N 151°0.38' E | 5404-5398 | 62 | |
| 10-3 | 25.08.2012 | 41°12.01' N 150°5.70' E | 41°12.19' N 150°6.40' E | 5249-5248 | 62 | |
| 11-3 | 28.08.2012 | 40°12.93' N 148°6.04' E | 40°12.92' N 148°5.41' E | 5348-5344 | 61 | |


**Table 2.** Lebensspuren and associated tracemakers identified in the present study. Note
that several lebensspuren can be produced by different tracemakers.

| Behaviour | Morphology | Description | Tracemaker taxonomy | Notes |
|---|---|---|---|---|
| Dwelling | Mounds | Large, smooth-sided cone structures. The diameter of the mounds ranged between 5 to 20 cm. | Unknown | Probably crustaceans |
| | Single burrows | Single entry holes within the flat sediment surface. Occasionally, a smooth, cone-shaped mound with a burrow entry hole at the apex. The diameters were varied, as large as 2 cm, but usually between 0.5 to 1 cm. | Unknown | |
| | Paired burrows | Two burrow entry holes that are closely spaced. The spacing between burrows was between 2 and 4 cm. | Bivalves and polychaetes | |
| | Cluster burrows | Three or more burrow entry holes that are closely and randomly spaced. The spacing between burrows was between 2 and 10 cm. | Unknown | Probably crustaceans |
| | Lined burrows | Three or more burrow entry holes that are aligned following a rectilinear or slightly sinuous pattern. | Unknown | Probably crustaceans |



| | | | | |
|---|---|---|---|---|
| | Crater cones | Large central mounds surrounded by distinctive clusters of round, shallow impressions. | Unknown | |
| Wasting | Crater | Depression holes related to the collapse of horizontal burrows | Actiniaria fam. gen. sp. 3 | Probably also other actiniarians |
| | Rounded faecal cast | Neat, short spirals of thick faecal matter | cf. *Elpidia*. sp. 1, *Scotoplanes* sp. 1, *Scotoplanes* sp. 2 | Due to image resolution, *Elpidia* rounded faecal casts (which are commonly <1cm in size) have only been recognized on a few occasions (when it was in focus) |
| | Smooth faecal cast | Smooth thick faecal matter with a straight or slightly sinuous shape. | cf. *Pseudostichopus* sp. *Psychropotes* morphospecies 1, *Psychropotes* morphospecies 2, *Synallactidae* morphospecies 1 (Amon et al. 2017), *Benthodytes* sp. 1 | Smooth faecal cast from *Benthodytes* sp. 1 may present compressed appearance. |
| | Mounded faecal cast | Discrete piles of faecal matter which are not associated with burrow entry holes. | Unknown | |
| | Coiled faecal cast | Thick faecal strings appearing compressed and curled with one straight coil at the end. May be present along thick trail lines. | *Psychropotes* morphospecies 1, *Psychropotes* morphospecies 2, *Benthodytes* sp. 1 | |
| | Knotted faecal cast | Tightly loop faecal trails, often with a characteristic loop-hook at the end. | *Peniagone* sp.1 to *Peniagone* sp. 3 | The bigger morphotypes of this faecal cast belong to *Benthodytes* sp. 1 |
| | Wavy faecal cast | Tiny (less than 0.5 cm in thickness) meandering faecal remains with variable length and often in fragmented form. | *Peniagone* sp.1 to *Peniagone* sp. 3 | Possibly formed by uncoiling of knotted faecal cast |
| | Switchbacks faecal cast | Switchback or meandering feature often beginning or ending in a spiral. The acorn worm is often observed making the feature. | *Torquaratoridae*. gen. sp. 1 | |
| | Spirals faecal cast | Faecal spirals with both clockwise and anti-clockwise paths. The acorn is often observed making the feature. | Enteropneusta gen. sp. 1, Enteropneusta gen. sp. 2 | |
| Locomotion and Feeding | Rosette-shape | Small burrow entry hole with thick, radial spokes from the central burrow. Partially completed rosettes are commonly observed. Spokes vary in thickness and length. Mounds are often found in close proximity to the rosette. | Unknown | This trace is usually related with echiuran worms but none has been observed in this study |
| | Thick M-trails | Complex concave crawling structures, ranging in width from 3 to 15 cm. Both sides of the trail have small sediment ridges (forming a M-shape trail) due to the movement of the tracemaker through the seafloor. The trails are straight and most commonly sinuous; occasionally observed with the echinoids forming the track. | Asteroidea. fam. gen. sp. 1, Asteroidea. fam. gen. sp. 4, Echinocrepis. sp. 1; Echinoidea. fam. gen. sp. 5 | |
| | Thick flat trails | Smooth concave trails of varying length with occasional small sediment puncture marks. Thickness ranges from 2 to 10 cm. Trails may form linear, meandering, or discontinuous paths. | cf. *Benthodytes* sp. 1, *Psychropotidae*, Asteroidea. fam. gen. sp. 3; Echinoidea. fam. gen. sp. 2; Echinoidea. fam. gen. sp. 7 | |
| | Thin trails | Smooth, concave trails of varying length, up to 2 cm thick. Trails may form linear, meandering or completely random paths | Gastropoda. fam. gen. sp. 1 to Gastropoda. fam. gen. sp. 6; Echinoidea. fam. gen. sp. 6 | |
| | Mounded trails | Smooth, with occasional ploughed features, convex trails of varying length and 3-10 cm thick. Trails may form linear, meandering or completely random paths. Craters appear sometimes in the middle of the trail. | Unknown | |
| Resting | Asteroid impressions | Asteroid star-shaped depressions with different dimensions. Diameter ranges from 1 to 15 cm. | Asteroidea. fam. gen. sp. 3, 4, 7, 8, 9 | |
| | Ophiuroid impressions | Ophiuroid star-shaped depressions | Ophiuroidea. fam. gen. sp. 1 to Ophiuroidea. fam. gen. sp. 3 | |





| Circular impressions | Circular depressions with a depth of less than 4cm | Actiniaria. fam. gen. sp. 1, Actiniaria. fam. gen. sp. 3, Actiniaria. fam. gen. sp. 7 |
|---|---|---|


**Table 3.** Total number of lebensspuren, tracemakers, degrading fauna and fauna

identified through the 8 deep-sea stations at the Kuril Kamchatka area.


| | | | Tracemakers | | | | | Lebensspuren | | | | |
|---|---|---|---|---|---|---|---|---|---|---|---|---|
| N=50 (frames per station) | **Fauna** | **Degrading fauna** | Total | *Wasting* | *Locomotion and feeding* | *Resting* | *Dwelling* | Total | *Wasting* | *Locomotion and feeding* | *Resting* | *Dwelling* |
| **Station 3** | 560 | 95 | 91 | 7 | 1 | 81 | X | 1207 | 63 | 84 | 361 | 699 |
| **Station 4** | 609 | 271 | 250 | 174 | 7 | 70 | X | 991 | 257 | 30 | 195 | 509 |
| **Station 5** | 157 | 27 | 20 | 11 | 10 | 7 | X | 974 | 557 | 18 | 37 | 361 |
| **Station 6** | 750 | 25 | 19 | 9 | 5 | 9 | X | 569 | 257 | 36 | 32 | 240 |
| **Station 8** | 522 | 52 | 36 | 3 | 6 | 27 | X | 321 | 77 | 32 | 32 | 178 |
| **Station 9** | 723 | 119 | 108 | 86 | 6 | 17 | X | 2448 | 2069 | 25 | 60 | 292 |
| **Station 10** | 181 | 32 | 13 | 5 | 8 | 4 | X | 687 | 278 | 46 | 27 | 328 |
| **Station 11** | 507 | 169 | 139 | 130 | 2 | 5 | X | 2229 | 1803 | 50 | 13 | 363 |