# Peer review of "Diversity and density relationships between lebensspuren and tracemaking organisms: a study case from abyssal Northwest Pacific"

_Biogeosciences, 2023_

## Author Response (AR1)

**Reviewer 1**

We really appreciate Dr. Przeslawski time and efforts to provide a detailed and quick review. We would like to thank the reviewer for their valuable and constructive comments that have been so helpful to improve the previous version of the manuscript. Here you can find a point by point response to all the raised comments.

The reviewers' comments are in italics and our responses with regular typing to make it easier for you to check.

*GENERAL COMMENTS*

*This study uses deep-sea imagery to investigate the relationships between lebensspuren (life traces on sediment) and various faunal groups, including those that make and those that degrade lebensspuren. The authors find high spatial variation and limited relationships between lebensspuren and faunal diversity and density. They relate their findings to previous similar studies in other deep-sea environments to try to understand broader patterns between lebensspuren and deep-sea diversity.*

*I find the ecological theory and introduction to this study a bit overstated and at times unclear. At its core, this study is a straightforward ecological analysis of lebensspuren and other biological and ecological variables. Its novelty is the differentiation of fauna into tracemakers and degrading fauna. The repeated emphasis on testing a 'diversity and density hypothesis' is vague and unclear. There are also odd phrasings used ('diversity and density hypothesis', 'tracemaker-lebensspuren tandem'), and the use of 'species' is incorrect.*

The reviewer is correct and the novelty of this study is to subdivide the fauna into tracemakers and degrading fauna for later diversity and density comparisons. We have modified the abstract and introduction (i.e., aims paragraph) to highlight this. Also, we added a last statement on the conclusions to highlight this.

The emphasis on "testing diversity and density hypothesis" is to inform the reader that different correlations have been observed previously between the fauna and lebensspuren (Kitchell et al., 1978, Young et al., 1985; Przeslawski et al., 2012). However, following your suggestion we have simplified the phrases and statements that expose these variable correlations.

We agree with the reviewer that tracemaker-lebensspuren tandem may confuse the reader. We have modified this.

The reviewer is correct on the use of "species". None of these studies treated the Lebensspuren as a species, which is why we had put it in quotes. With the phrase we wanted to indicate that previous studies treated the lebensspuren morphotypes as "species" for statistical treatments (accumulation curves, diversity indices, etc). However, we think that the reviewer is right and the use of "species" is incorrect and can lead to confusion (see answer above for response to general comments). Therefore, we have modified the sentence.

Also, we decided to use the term "taxon" (for the benthic fauna) instead as this is more generally valid, including morphologically discriminated species as well as higher levels of the systematic hierarchy, which in some cases had to suffice, e.g., due to poor image quality. The appropriate paragraph in the methods section has been adapted accordingly.

*The discussion is well-written and goes into detail about the nuance and complexity of the ecology of lebensspuren in deep-sea environments. The authors use an old lebensspuren classification system from 1954 when there is a more recent one that modern studies use (see Althaus et al 2015, Przeslawski et al 2012). It would be easier to relate the data and results here to previous studies if the authors used the same classification system as them. I suggest re-doing the classification and analyses (or at the very least, justifying why it was used and showing how it maps to the modern lebensspuren classification system).*

The classification we used is based on the two most relevant pillars of ichnology and have been adapted so they can usefully be applied to the data source we studied – still images. These pillars are: 1) behaviour (simplified Seilacherian categories; i.e., adapted to be applied to still images) and 2) morphology (morphology as the main ichnotaxobase; Bertling et al., 2022). However, being well aware of the published classification schemes the reviewer mentioned in regard to morphological names (Dundas and Przeslawski 2009, Przeslawski et al 2012, Althaus et al 2015) and to avoid unnecessary confusion and problems the same morphotype names have been applied – in most cases- in our study where possible without creating synonyms are ambiguous categories. Thus, our classification takes into account the state-of-the-art and widely accepted classification schemes that and, hence, we think that re-doing the classification is not necessary nor the analyses. Moreover, the diversity and density data will not vary due to just changing the names of the lebensspuren morphotypes. For more detail the classification we used is explained as follows:

The first one is related with the inferred tracemaker behaviour. In his pioneering paper Seilacher (1954) described the following behaviour in relation with lebensspuren: i. Resting lebensspuren—imprints of stationary animals. ii. Crawling lebensspuren—displaced sediment by movement of deposit feeders, sometimes marked by depressions left by the limbs. iii. Feeding structures—faecal casts and pellets. iv. Grazing lebensspuren—minor/fragile disturbances to sediment surface. v. Dwellings—mounds and burrows. However, these categories are rather difficult to assess from still images (e.g., differentiating between crawling and grazing or feeding vs. grazing). For example, in Althaus et al (2015) trails are assigned to crawling. The original crawling definition (Seilacher, 1954; Frey, 1973): trackways and trails (epistratal or intrastratal) made by animals traveling from one place to another. We know that some of these trails are carried out by echinoid that not only travel but also feed at the same time (e.g., Miguez-Salas et al., 2022). Thus, a crawling assignation to trails is not the most appropriate one.

The behavioural classification here is part of a neoichnotaxonomical manuscript that we submitted to another journal in collaboration with many ichnotaxonomical colleagues which differentiates 5 behaviours that can be identified in still images: Resting (cubichnia), Locomotion (repichnia), Dwelling (domichnia), Feeding (fodinichnia), and Wasting (digestichnia). Finally, with respect to this manuscript and the articles cited by the reviewer (Dundas and Przeslawski 2009, Przeslawski et al 2012, Althaus et al 2015), these papers use

the waste cast category for faecal cast which are the most abundant lebensspuren in our study. However, the "waste cast" category is not a behavioural category in ichnological research. Even though it is not original seilacherian ethological category, wasting (digestichnia) is the proper name for this behavioural category which in general terms refers to "Wasting traces are formed as the organism excretes sediment particles from which organic material has been absorbed" (see Vallon et al., 2015 "An updated classification of animal behaviour preserved in substrates"; Bertling et al., 2022 "Names for trace fossils 2.0: theory and practice in ichnotaxonomy").

The second one relates to the lebensspuren morphology. For this classification in fact we have tried to keep, as much as possible, the names proposed in Dundas and Przeslawski 2009, Przeslawski et al 2012, Althaus et al 2015. We have added some sentences in the methodology to highlight this. In fact, in the case of the wasting, resting and dwelling lebensspuren, most of the names are the same, following the proposals in the mentioned papers. We only modified the locomotion-feeding lebensspuren names because previously published names are rather simplistic (thick or thin trail) and do not offer any information about the lebensspuren seafloor epirelief. Since Frey (1973) wrote "Concepts in the study of biogenic sedimentary structures" until the recent ichnotaxonomical review of Bertling et al. (2022), one statement has remained, the morphology of the trace should be the corner stone and the main ichnotaxobase. Taking this into consideration, when describing a biogenic trace, the relief (see Fig. 4 in Frey 1973) is a basic morphological feature. That is why we have preferred to add flat or M-trail (when you have parallel sediment ridges in the margins) in order to give further detail in trail classification (rather than just saying thick or thin). This is also that we also discuss in the submitted neoichnotaxonomical manuscript.

*SPECIFIC COMMENTS*

*The title is not readily understood most scientists, as the 'diversity and density dilemma' and 'lebensspuren-tracemaker tandem' are not common terms and border on jargon. Please rephrase to something more meaningful to a broader readership. E.g. 'Relationships between lebensspuren and tracemaking organisms: a study case from the abyssal Northwest Pacific'*

We agree with the reviewer; the title is perhaps too technical for a more general audience. Therefore we have modified it following her suggestion.

*Line 18: change 'bioturbational processes' to 'bioturbation'*

Done

*Line 29: 'between specific stations' doesn't make sense. Do you mean 'at specific stations'?*

The reviewer is right. We have modified it.

*Lines 37-42: There are too many statements in this last concluding sentence. Summarise more succinctly, or break into smaller sentences.*

We have separated the statements.

*Line 45: change 'as well as' to 'in relation to'. The study of animals and their substrates is not neoichnology; the study of traces of extant organisms is neoichnology.*

The reviewer is right. We have restructured the sentence.

*Line 44 and elsewhere: 'change 'bioturbation processes' to 'bioturbation'.*

Done

*Line 47: change wording. Lebensspuren are not 'highly precise portraits' of ecology. Their utility as proxies of biodiversity and ecological relationships remains uncertain and likely varies among ecosystems and species…as this study shows.*

We have removed the term "highly precise" to not overstate the sentence

*Line 65: This is incorrect - neither of these studies referred to lebensspuren morphotypes as species.*

The reviewer is correct. See above response.

*Lines 95-97: Variations in study findings does not necessarily mean a field is in its infancy. Other possibilities: Scale is too coarse to detect relationships in some systems and datasets- the nature of lebensspuren means that one is often operating a much coarser taxonomic resolution than organism identification. 2) ecological relationships vary across regions and ecosystems – not all abyssal environments are the same. You describe some of this well in your discussion.*

The reviewer is right - variability does not mean that a field is in its infancy. We modified that to just expose that there are few detailed lebensspuren studies to emphasize that it is a field that needs further exploration.

*Line 106: The 'one step further' is not clear here. The novelty of this study is the differentiation of marine fauna into 'tracemakers' which should be positively correlated to lebensspuren and 'degrading fauna' which should be negatively. The attempt to relate to broader ecological theory (e.g. 'previous diversity and density hypotheses…') is vague and confusing.*

We have modified the objectives and how the explanation on how this study goes "one step further".

*Line 151-155: Why did you use the 1954 classification system, and how does this map to the one used in much more recent lebensspuren studies (e.g. Bell et al, Przeslawski et al) and classification systems (e.g. CATAMI in Althaus et al 2015)?*

See above response.

*Line 164-165, 172-173: Don't do this. A 'morphotype' is not the same as 'species'. Use 'morphotype' for lebensspuren and 'morphospecies' for fauna throughout. It is just as simple and is correct.*

We have modified this throughout the manuscript. However, we decided to use the term "taxon" instead as this is more generally valid, including morphologically discriminated species as well as higher levels of the systematic hierarchy, which in some cases had to suffice, e.g., due to poor image quality. The appropriate paragraph in the methods section has been adapted accordingly.

*Line 214-216: What does this difference in standard deviation mean? That faunal diversity was more consistent among sites that the other diversity indices? Provide some context to these statements please.*

It means what the reviewer suggests. We have added this statement following her suggestion.

*Lines 217-218: '... with COMPARABLE DIVERSITY INDICES FROM the other three groups…'*

Rephrased

*Line 223: 'Wasting lebensspuren' isn't great terminology since 'wasting' is an adjective that does not fit in this case – use 'waste lebensspuren'*

Done

*Line 229: reword 'Lebensspuren assemblages were generally similar among station (Fig 5A).'*

Done

*Line 234-241, Figure 2: Please use the same terms as previous studies for lebensspuren morphotypes – it looks like most of what you're preferring to has already been classified in Przeslawski et al 2012.*

See above response about lebensspuren naming concerns. The reviewer is correct and in the manuscript we have tried to stick to previously proposed names as much as possible (obtained from: Dundas and Przeslawski, 2009; Przeslawski et al 2012; Althaus et al., 2015)

*Line 251: change 'the obtained results' to 'results'*

Done

*Line 252: What hypotheses or studies are you referring to? Recent studies suggest that they are indeed complicated relationships. This study supports this as well.*

The reviewer is right. We added the references and modified the sentence.

*Line 284: I argue that unknown lebensspuren are an equal challenge – there are still some traces for which we have absolutely no idea what made them. See linear holes in Veccione et al 2022 (https://www.frontiersin.org/articles/10.3389/fmars.2022.812915/full) and spider trace in Przeslawski 2022 (https://www.frontiersin.org/articles/10.3389/fmars.2022.1086193/full)*

The reviewer is right. We modified the examples in the sentence. We recently publish a paper (Brand et al., 2023) in relation to Veccione et al., 2022's mysterious sublinear burrows. We found "similar" lebensspuren and one possible tracemaker (which does not mean that the other linear burrows are produced by amphipods as well). But, we know that for many dwelling lebensspuren an unknown classification is the most plausible one…..ultimately affecting diversity comparisons (we accordingly modified Line 336).

*Line 311: Has there been any attempt with automated image recognition for lebensspuren? I'd be keen to have a sentence or two discussing how AI may assist these kind of imagery analyses in the future.*

Yes indeed. This is something I discussed with Jen Durden and colleagues as well as within the FathomNet framework. The use of AI is promising for benthic fauna recognition. However, in the case of lebensspuren the future is more discouraging for a simple reason: the colour of the lebensspuren is almost always the same as that of the seafloor (background colour) therefore the algorithms cannot differentiate them. I tried personally one algorithm that was recently published (https://github.com/dmair1989/imagegrains) to quantify dimensions of river grains in planar surfaces – with the idea of identifying faecal cast automatically - and the result was not good at all. Mainly because the lack of colour contrast. I added a sentence explaining this limitation.

*Line 324-328: repetitive*

We agree. We deleted the first sentences to avoid repetition.

*Line 393-94: delete 'it has to be kept in mind' and other similar filler phrases throughout the ms*

Done

*Lines 395-96: citation needed for these previous studies*

Done

*Table 2: How does this relate to existing lebensspuren catalogue (Przeslawski et al 2012, Dundas and Przeslawski 2009)?*

See above response for lebensspuren naming concerns.

*TECHNICAL COMMENTS*

*Line 412: reliably*

Modified

*Figure 4: subdivided*

Modified

**Reviewer 2**

We would like to thank Dr. Purser for his valuable and constructive comments that have been so helpful to improve the previous version of the manuscript. Here you can find a point by point response to all the raised comments.

The reviewers' comments are in italics and our responses with regular typing to make it easier for you to check.

*General Comments:*

*Kuril Kamchatka Trench image data – where is this? This study uses data collected by the German research vessel Sonne, though it does not give any publically accessible link to the collected image data, video data or described extracted image frames from which all the statistical investigations are based. This is not appropriate I believe for a current publication at this level, and making such data publicly available is a requirement for the current Sonne data I believe. Making this data available is essential to consider the results. I believe since the analysis was carried out with BIIGLE 2.0, the output of the analysis should (or in this case, could, NOT essential) be also uploaded to a public repository, either as a file associated with a raw data PANGAEA upload for example, or to a pure data repository such as MENDELEY DATA.*

We totally agree with the reviewer.

All still images retrieved from the OFOS videos have been uploaded to Zenodo (i.e., more than 2500 still images among all deep-sea stations), making such data publicly available.
      Miguez Salas, O., & Riehl, T. (2023). Still images from the KuramBio expedition 2012 (Stations 3-6, 8-11) obtained with the Ocean Floor Observation System. Zenodo. https://doi.org/10.5281/zenodo.10057539.

Additionally, the annotation reports generated with BIIGLE 2.0 for the lebensspuren and benthic fauna in each station was uploaded as an excel dataset to Zenodo. In this way, the data on which we have based our analysis is made public. In the first column of the Excel the codeframe is indicated (i.e., the still image in which the annotations were made)
      Miguez Salas, O., & Riehl, T. (2023). Lebensspuren and benthic fauna diversity and density data obtained from KuramBio 2012 expedition still images (50 still images per 8 deep-sea stations) [Data set]. Zenodo. https://doi.org/10.5281/zenodo.10057636.

We add both information and references to the manuscript in the methodological section.

*Line 44 on      This section introduces Neoichnology but does not state clearly how the link between animals and substrates is made. Is it imaged based , video based etc? This needs to be clearer. IN line 75 'improvements in imaging technology' is first mentioned, but the fact that images are used should be introduced earlier, with introduction of concepts such as illumination, viewing angles etc.*
Neoichnology studies the interactions between animals and substrates (e.g., bioturbation, bioerosion) in modern environments. These studies have a wide spectrum going from in-situ

measurements (e.g., morphological measurements with burrow resin casts), tank experiments, etc. However, we added a sentence to clarify that lebensspuren studies, in the marine realm, are mostly image-base.

*Discussion      English is a bit disjointed, some points addressed in the detailed response, but this section in particular needs an English native full readthrough to smooth off all edges. The list of comments on grammar in the detailed responses should NOT be considered total.* The manuscript has been reviewed by an English native speaker from our research group and minor changes have been made throughout the manuscript.

*Particularly in the Conclusions, there is a tendency to overextend the findings from the current study to interactions between fauna and fauna traces globally. The paper should be focused more specifically on discussing the assessed data relevance locally, though of course commenting on potential global relevance of approach and observations.*
We modified the term global on the conclusion section.

*I think the statistical analysis and discussions of the findings are interesting, well presented generally (save for some grammatical awkwardnesses) and placed into context with appropriate literature. I firmly believe though that presenting the summary output based on images not publically available as a supplementary file is insufficient to be fully convinced and engaged in these analyses.*
See above comment on the creation of a public repository with all the still images used for this analysis.

*I think the authors can address these points readily and i look forward to citing a revision of this study soon!*

*Here below, some detailed comments by line number:*

**Detailed Comments:**
All the detailed comments has been accepted and modified in the text.

*Line 9          Seafloor landscapes to landscapes*
Done
*Line 33           maybe make more explicit with 'total' faunal densities as next sentence links densities with specific fauna type (i.e. tracemakers)*
Done
*Line 41          depending to depending on*
Done

*Line 47          precise portrait seems excessive, perhaps, 'lebensspuren' abundances can clearly relate environmental conditions to animal responses… or something like this.*
Done

*Line 87           seemed to seems*
Done

*Line 87 – 89      perhaps these contrasts are regional or ecosystem specific, and not a general and overriding difference?*
Done

*Line 102      misspelling diversity*
Done

*Line 140 – 144   Where are these extracted frames visible? How can the results be checked if not publicly available? How about also making the BIIGLE 2.0 output available via a supplementary file or via an online repository?*
See above response

*Line 296      morphotypes to morphotype.*
Done

*Line 241      misspelling Asteroidea*
Done

*Line 312      may allow to improve to may allow improvement of*
Done

*Line 314      will allow to characterize to will allow researchers to characterize*
Done

*Line 317      than what could to than could be achieved within this study*
Done

*Line 320      different fauna so predominantly endobenthic lifestyleS*
We do not know what the reviewer means.

*Line 412      misspelling reliably*
Done

*Line 416      may be control to may be controlling*
Done